# Mechanism and Therapeutic Targets of c-Jun-N-Terminal Kinases Activation in Nonalcoholic Fatty Liver Disease

**DOI:** 10.3390/biomedicines10082035

**Published:** 2022-08-20

**Authors:** Robert W. M. Min, Filbert W. M. Aung, Bryant Liu, Aliza Arya, Sanda Win

**Affiliations:** 1Rush Medical College, Rush University, Chicago, IL 60612, USA; 2Brown University, Providence, RI 02912, USA; 3Division of Gastrointestinal and Liver Disease, Department of Medicine, Keck School of Medicine, University of Southern California, 2011 Zonal Ave., HMR 612, Los Angeles, CA 90089, USA

**Keywords:** JNK, SAB, JNK activation loop, ROS, MAP kinase cascade, drug therapy, non-alcoholic steatohepatitis

## Abstract

Non-alcoholic fatty liver (NAFL) is the most common chronic liver disease. Activation of mitogen-activated kinases (MAPK) cascade, which leads to c-Jun N-terminal kinase (JNK) activation occurs in the liver in response to the nutritional and metabolic stress. The aberrant activation of MAPKs, especially c-Jun-N-terminal kinases (JNKs), leads to unwanted genetic and epi-genetic modifications in addition to the metabolic stress adaptation in hepatocytes. A mechanism of sustained P-JNK activation was identified in acute and chronic liver diseases, suggesting an important role of aberrant JNK activation in NASH. Therefore, modulation of JNK activation, rather than targeting JNK protein levels, is a plausible therapeutic application for the treatment of chronic liver disease.

## 1. Introduction

Non-alcoholic fatty liver (NAFL) is a disorder where excess fat accumulates in the liver (steatosis) due to non-alcoholic or non-viral causes. Non-alcoholic steatohepatitis (NASH) is hepatic steatosis associated with hepatocellular injury, innate immune cell-mediated inflammation, and progressive fibrosis in the liver. A total of 30–40% of adults in the United States have NAFL, defined as hepatic fat >5% of total liver weight, and 3–12% have NASH. Twenty percent of people with NAFL will develop NASH. NASH can progress to irreversible liver diseases such as cirrhosis and hepatocellular carcinoma (HCC). Up to 25% of adults with NASH may have cirrhosis [1,2,3,4,5,6]. While liver transplant is the ultimate solution for cirrhosis or HCC, new drugs are emerging that target molecules and pathways such as stress kinases, *de novo* lipogenesis, lipid oxidation and transport, lipotoxicity, inflammation and fibrogenesis to prevent or treat steatosis and steatohepatitis in liver [4,7,8,9]. Hypertension and cardiovascular diseases (CVD) due to increased risk of arteriosclerosis, hyperglycemia due to hepatic gluconeogenesis, and hyperinsulinemia due to insulin resistance cause the metabolic syndrome in NAFL disease [1,3]. Therefore, an effective therapeutic management is required for NAFL disease to prevent progression to NASH and its resulting complications. While the mechanism of developing NAFL/NASH is not fully understood, the hepatic metabolic stress response via JNK activation has been identified as a common pathway in numerous models of liver injury [10,11,12,13,14,15]. The relationship of JNK activation pathway to oxidative stress, lipotoxic stress and cell death, inflammation, and cytokines, fibrogenesis, *de novo* lipogenesis, lipolysis, lipid oxidation and transport are being studied to determine the mechanistic significant of MAPKs in the development of NASH (Figure 1) [6,16,17,18,19,20]. In addition, the development of NAFL/NASH is influenced by many common diseases whose primary etiology is not in liver, such as type II diabetes, hyper-insulinemia, obesity, changes in sex hormone regulation, adipokines and adipose tissue lipolysis, dysfunctional muscle metabolism, bile acid dyshomeostasis, gut dysbiosis, and brain lesions, and those are beyond the scope of this review [21,22,23,24,25,26,27,28,29]. This review discusses the hepatic stress kinase activation in NASH and plausible pharmaceutical targets.

## 2. Hepatic MAPK Family in Metabolic Stress and the Mechanism of Sustained Activation

The liver is composed of 60% parenchymal cells, i.e., hepatocytes, and 30% to 35% non-parenchymal cells, i.e., Kupffer cells (KCs), hepatic stellate cells (HSCs) and liver sinusoidal endothelial cells (LSECs). Hepatocytes are the work horse of the liver and carry out a vast array of metabolic, regulatory, and toxicological functions. Hepatocytes express MAPKs which transduce extracellular and intracellular signals to regulate cell proliferation, differentiation, apoptosis, and metabolism [30]. Metabolic stress induced MAPK cascade activation in hepatocytes includes an upstream MAPK kinase kinases (MAP3K) such as ASK1, MLKs, TAK1, MAPK kinases (MAP2K) such as MKK4, MKK7, and terminal stress kinases such as JNK, and p38 [31,32,33,34]. Diets, specifically hyper-nutrition type diets such as high fat, high carbohydrate, activate hepatic JNK longer than standard fat and carbohydrate diet [35]. It is important to mention that activated JNK (P-JNK) level decreases between meals, and fasting overnight alleviates basal low level of P-JNK. Hyper-nutrition type diets activate JNK by metabolites, endoplasmic reticulum (ER) stress, and mitochondrial stress via the modification of acetylation/deacetylation status, transcriptional activation/inhibition, energy metabolism and lipid oxidation, and oxidative stress [36,37,38].

JNK and p38 stress kinase are self-regulated via dual-specificity phosphatase (DUSP) group, which is the vital mechanism to dampen and terminate kinase activation via direct inactivation or inhibiting upstream kinase activation [39,40]. Individual DUSP has different tissue expression, subcellular localization, and has differing preferences for the kinases in MAPK cascade such as DUSP22 in ASK1-MKK7-JNK signaling pathway, DUSP5 on ERK signaling pathway [41,42,43]. DUSP9, 10, 12, 14 and 26 in liver have been found to prevent the hepatic steatosis [44,45,46,47,48,49,50]. In addition, the scaffold function of DUSP has been noted and extended reviews on DUSPs are available for further reading [41].

The sustained activation of JNK via the self-amplified feed-forward activation loop has been discovered recently [51] and found to play a key role in several disease models such as drug induced acute liver injury and cell death, cytokine induced hepatic apoptosis, lipotoxicity, ER stress induced mitochondrial dysfunction and cell death, ischemia-reperfusion injury in heart, cardiotoxicity, neuronal activity in brain, neurotoxicity, and ovarian cancer treatment [52,53,54,55,56,57,58,59]. Initial activated P-JNKs translocate to mitochondria where P-JNKs phosphorylate mitochondrial outer membrane protein SH3 homology associated BTK binding protein, SH3BP5 (SAB). P-JNK binding to SAB initiates intramitochondrial release of tyrosine phosphatase SH2 phosphatase 1 (Shp1) from SAB leading to dephosphorylation of activated-Src at tyrosine 419, which occurs on and requires the platform, docking protein 4 (DOK4), located on the mitochondrial inner membrane. P-Src is required to maintain electron transport [51]. Decreased P-Src inhibits mitochondrial respiration and enhances reactive oxygen species (ROS) production especially when mitochondrial aerobic metabolism via tricarboxylic acid cycle (TCA cycle) is upregulated by Ca^+2^ flux from mitochondria-associated membrane (MAM), ER and cytoplasm (Figure 1 and Figure 2) [54]. ROS facilitates ASK1 N-terminal dimerization and activation by oxidation and removal of thioredoxin, which binds and inhibits ASK1 [60]. Activated ASK1 activates MKK4/7 which in turn activates JNKs to amplify and increase the level of P-JNK [30,61,62]. Interestingly, P-MKK4 is associated with P-JNK on mitochondria [63]. Importantly, SAB is a pivotal molecule determining the level of P-JNK to cellular stress and damage [51,64]. Depletion or inhibition of ASK1, MKK4 or SAB prevents sustained P-JNK activation and prevents JNK mediated cellular stress but does not prevent initial JNK activation from the stress inducers such as reactive adducts in toxic drug metabolites, metabolic reprogram, IRE1α and PERK activation in unfolded protein or ER stress, membrane associated Src activated in lipid induced modification of plasma membrane plasticity and fluidity [52,65,66,67,68]. Therefore, the feed-forward activation to sustain JNK activation, the JNK-SAB-ROS activation loop, is a key player in the mechanism of cellular stress and damage [30,51].

The JNK-interacting protein (JIP) family plays an integral role in MAPK cascade activation by creating proximity of upstream kinases and terminal kinases [69,70]. JIP1 is a key platform in MLKs mediated MKK7 and JNKs activation [71,72]. However, JIP1 also mediates ASK1-MEK-JNK signaling [73] and more than one JIP could be involved in P-JNK inactivation and activation loop [74,75]. We believe MLKs activation is an important signal for initial JNK activation at least in the acute liver injury models because initial JNK activation occur in the liver of acetaminophen treated ASK1 KO mice [76]. Role and function of JIP2, 3 and 4 are yet to be examined. Stress kinases target various intracellular molecules such as transcriptional and non-transcriptional targets via kinase interacting motif (KIM) and substrates are predicted to be phosphorylated serine/threonine-proline (SP) sites [30,61,62]. As an example, JNK targeted GCLC are ubiquitinated and degraded leading to decreased GSH level and increased oxidative stress as occur in the acute acetaminophen hepatotoxicity [64,77]. In NASH, P-JNK targeting of transcriptional factors, such as SREBFs to increase lipogenesis and cholesterol synthesis, NCoR1 to repress PPARα and fatty acid oxidation, leads to cumulative lipotoxic stress [78,79,80]. Therefore, P-JNK-SAB ROS activation loop is a key mediator in sustained activation of JNK.

## 3. Hepatic MAPK Mediates Progression of NAFL/NASH

### 3.1. Hepatocytes: Hepatic Steatosis, Oxidative Stress, and Autophagy

Hepatic fat is mostly triglyceride formed by esterification of free fatty acid from the diet, adipose tissue lipolysis and *de novo* lipogenesis. Fat accumulation in hepatocytes causes lipotoxicity and apoptosis [6,16,53] and ballooning degeneration of hepatocytes which are associated with oxidative stress in steatosis/steatohepatitis [78]. In the course of disease progression, high calorie diet increases mitochondrial function and metabolism accompanied with stress kinase activation [78,81]. Interaction of activated P-JNK with mitochondrial protein SAB impairs mitochondrial respiration and produces ROS leading to amplify feed-forward activation of JNK, known as the JNK-SAB-ROS activation loop [30,51,82]. Decreased hepatic SAB (SH3BP5) expression by *Sab* knockdown or knockout decreases ROS and decreases the stress kinase JNK activation in established experimental models of steatosis and steatohepatitis [78]. Notably, increased expression of SAB in steatotic liver plays a pivotal role in the higher activation of stress kinases and progression to steatohepatitis [64,78]. In line with this observation, depletion of hepatic JNK1/2 prevents diet induced lipogenesis and steatosis [79]. In addition, WW domain containing transcription regulator 1 (Wwtr1/TAZ) induced hepatic NADPH oxidase 2 (NOX2/*Cybb*) expression mediates the oxidative DNA damage in diet induced NASH and HCC [7]. NOX2 is a superoxide generating enzyme, which delivers activated oxygen into phagocytic vacuole in inflammatory cells such as granulocytic neutrophil in NASH [83]. The mechanism may be associated with microbiome dysregulation in the gut and translocation to liver [84]. Nevertheless, the inter-regulation of MAPK family and TAZ expression and activity in the progression of NASH needs to be explored.

The critical role of JNK activation in the progression of NASH is supported by studies of MAP3Ks such as ASK1 and MLKs, and DUSPs. For instance, *ASK1^−/−^* mice have reduced hepatic steatosis. CASP8 and a FADD-like apoptosis regulator (CFLAR) disrupts the N-terminus-mediated dimerization of ASK1 and favors ASK1 degradation and prevents JNK activation and hepatic steatosis [85]. CFLAR peptide is proposed as a potential therapeutic agent. In addition, upregulation of TRAF1 promotes hepatic steatosis through enhanced activation of ASK1-mediated JNK/P38 activation [86]. However, contradictory data is reported in hepatocyte specific ASK1 deleted mice which is cross-bred with albumin-Cre mice or in mice treated with ASK1 inhibitor. Deletion of hepatic ASK1 decreases P-JNK/P-38, impairs autophagosome formation in the liver and increase the liver triglyceride and fibrosis [87]. Hepatocyte specific ASK1 expression in mice prevents hepatic steatosis [87]. It should be noted that the content of diet, age and background strain of mice, housing environment such as temperature and microbiome affect the animal model of NASH [88], and direct or indirect effect of ASK1 and CFLAR on NASH is required to reconcile. MLK2 and 3 are redundant and abundant in the liver. *MLK^−/−^* mice prevents high fat diet (HFD) induced hepatic steatosis and triglyceride accumulation. The mechanism is explained by reduced JNK/p38 activation in HFD fed *MLK^−/−^* mice, and in mice treated with MLK3 inhibitor URMC-099 [89]. However, further studies are required to examine the possible activation of ASK1 in HFD fed MLK KO mice, and vice versa to identify the specific or overlapping function in progression to NASH. Nevertheless, JNK activation is an important mechanism in development of hepatic steatosis.

The liver expresses several dual-specificity phosphatases (DUSPs). DUSP9 and 12 have the substrate preference for JNK and p38 [41]. Moreover, the expression of DUSPs protects against steatohepatitis through suppression of ASK1 activation [44,45]. The suppression of ASK1 activation could be via inhibition of JNK-SAB-ROS activation in DUSP9 and 12 over-expressed HFD fed mice. In line with these results, *DUSP10(MKP5)^−/−^* mice progress to severe steatosis with aging or HFD [46]. DUSP14 and 26 prevent JNK/p38 activation in hepatic steatosis and inflammation via inhibition of TAK1 [47,48]. In addition, DUSP14 protects against hepatic ischemia-reperfusion injury via suppressing TAK1 and subsequent NF-κB signaling and JNK activation [49]. The results of HFD fed *DUSP1^−/−^* mice are difficult to reconcile. Hepatic *DUSP1^−/−^* mice are protected from hepatic steatosis but have elevated JNK/p38 phosphorylation [90].

The dual role of JNK1/2 in HFD induced steatosis and steatohepatitis has been elucidated hepatic specific JNK1/2 depletion studies [79] and settled the earlier studies using gene specific KO mice [91]. Increased JNK activation with the progression of hepatic steatosis reduces peroxisome β-oxidation by suppressing peroxisome proliferator-activated receptor α (PPARα) activity, in part through the upregulation of nuclear receptor corepressor 1 (NCoR1) [79]. Moreover, the hepatic metabolic stress accompanied by increased JNK activation and increased flux of acetyl-CoA from mitochondria to cytosol favors acetylation of transcription factors and histones leading to *de novo* lipogenesis via transcriptional factors SREBFs and the activation of downstream lipogenesis genes [92]. In addition, the carbohydrate sensor ChREBP is upregulated following ingestion of a carbohydrate-enriched meal, leading to the increased expression of genes regulating lipogenesis and fatty acid esterification that promote liver steatosis [93]. Direct association of JNK activity and ChREBP expression and activation is unknown. P38 isoforms (α, β, γ/δ) expression and activity are difficult to reconcile in several studies of NASH. A recent study using liver-specific *p38α* KO mice suggested that hepatic p38α protects mice from steatohepatitis in a diet feeding model [94]. However, macrophage p38α promotes progression of steatohepatitis [95]. It is important to stress that pharmaceutical inhibition or knockdown of p38 causes liver injury [96,97] and isoform specific and conditional studies may be required to examine the hypothesis.

Autophagy, induced by extra- and intra-cellular stress, is the important part of liver homeostasis by removing damaged organelles (mitophagy, ER-phagy, pexo-phagy), lipid droplets (lipophagy) and protein aggregates, and recycling nutrients to preserve the cellular energy, integrity, and survival [98,99]. The mammalian target of rapamycin (mTOR) is the one major inhibitor of autophagy [100]. Growth factors and insulin repress autophagy via PI-3K-AKT-mTOR activation pathway [101,102]. During starvation, specifically glucose or amino acid deprivation, autophagy pathway in liver is upregulated by interfering mTOR activation via activation of cAMP-AMPK-TSC1/2 inhibitory pathway [101,102]. In diet-induced NASH, autophagy, especially lipophagy, occurs [103,104,105,106]. Increased lipophagy in NASH may reflect decreased mTOR activity due to P-JNK inhibition of insulin receptor signaling in NASH. However, the direct involvement of the P-JNK and P-JNK activation loop in a lipophagy mechanism is unknown. Nevertheless, pharmaceutical enhancement of autophagy by rapamycin or carbamazepine which inhibits mTOR reduced hepatic steatosis in NASH model [107], and severe hepatic steatosis occurs in mice with hepatic deletion of *Atg5* [104]. Viral expression of *Atg7* increases autophagy, reduces ER stress, and improves insulin sensitivity [108]. More importantly, ER stress in NASH may upregulate autophagy pathway because tunicamycin induces autophagy and IRE1α-JNK pathway is involved [109,110]. Though, JNK1 promotes autophagy through phosphorylation of BCL2, which releases BECN1 from BCL2 to contribute in autophagosome [111], JNK1 can phosphorylate RPTOR at Ser863 which is necessary to assemble MTOTC1 complex to block autophagy [112]. Direct evidence that JNK mediates these effects is needed to fully prove in NASH model.

Sex differences also exist in NAFLD [113]. The prevalence and severity of NAFLD are higher in men than in women during the reproductive age (age ≤ 50–60 years); however, women after menopause (age ≥ 50–60 years) have higher rates of NAFLD [113]. The hepatic ERα-p53-miR34a signaling axis decreases SAB protein level in women of reproductive age [64]. This is explained in hormone mediated suppression of SAB. SAB expression is reduced in adult females, compared to male and postmenopausal females [64]. Increased hepatic SAB expression increases JNK activation in steatosis and steatohepatitis [78]. Additional factors such as sex difference in metabolic homeostasis contributed by skeletal muscle, adipose tissue, and thyroid hormones [114,115,116] are beyond the scope of current review. Clinical and epidemiological studies have shown that postmenopausal women receiving hormone replacement therapy had a lower prevalence of NAFLD compared to postmenopausal women not receiving replacement therapy [113]. While these findings suggest that estrogen is protective, further studies are needed to define the hepatic molecular mechanisms of sex difference in NAFLD. Sex, age, reproductive status, and synthetic hormone usage are needed to include in future clinical investigation and gene association studies of NAFLD [113].

A meta-analysis of recent studies has also shown that the PNPLA3, also known as adiponutrin, rs738409 [G] allele is associated with an increased risk of diet related hepatocellular carcinoma [117]. The PNPLA3 Ile148Met variant is resistant to degradation and disrupts ATGL lipolysis activity [118]. In addition, levels of PNPLA3 and CGI-58 in lipid droplets determine the lipolysis activity of ATGL. A recent study demonstrated that cJUN inhibits RORα-mediated PNPLA3 expression [119], leading to downregulation of the ATGL activity. cJUN is a JNK activated transcriptional factor [62]. Thus, the relationship between hepatic stress kinase activity and hepatic lipase expression and activity needs to be further explored. An allele in PNPLA3 (rs738409[G], encoding Ile148Met) is associated with increased liver fat, hepatic inflammation, and fibrosis [120].

P-JNK activates transcriptional co-repressor NCoR1 and suppresses PPARα activation in NASH, affecting decreased expression of genes involved in fatty acid transport and oxidative degradation in mitochondria and peroxisomes [79]. Consistent with the hepatocyte-specific JNK deletion model, Elafibranor (GFT505, Genfit), a PPARα/β agonist, Saroglitazar, a PPARα/γ agonist, and lanifibranor, a pan-PPAR agonist normalize serum lipid profiles, insulin resistance and improve NASH [121]. PPAR agonists, one of the most advanced classes of anti-NASH molecules, are in phase II or III clinical studies and may need to improve the efficacy and safety [122]. In addition, long-term studies in rodents showed an association of PPARα agonists with hepatic carcinogenesis and the result could be species-specific effect [123,124]. In line with this observation, hepatic JNK KO increases the cholangiocyte proliferation, and intrahepatic cholangiocarcinoma [125]. JNK KO activates the transcription factor PPARα and its target genes related to hepatic cholesterol and bile acid synthesis resulting in cholestasis. Therefore, a pharmaceutical target that directly interferes and dampens the sustained P-JNK activation is required to overcome the effect of complete deletion of JNK.

### 3.2. Non-Hepatocytes: Immune Cells, Sinusoidal Endothelial Cells, Hepatic Stellate Cells

NASH is the hepatic steatosis with liver cell inflammation and innate-immune system activation involving Kupffer cells (KC) and production of pro-inflammatory cytokines in the liver. Extracellular vesicles, fatty acids, cholesterol released from steatotic hepatocytes and lipopolysaccharides (LPS) activate KCs and hepatic stellate cells (HSCs) via damage-associated molecular pattern receptors, known as Toll-like receptor 4 (TLR4) and TNF-related apoptosis-inducing ligand death receptor (TRAIL-R2) [126]. Mice without TLR4 in KCs are protected against steatosis and NAFLD progression [126]. KCs activated by the hepatocyte-derived mitochondrial DNA produce pro-inflammatory chemokines such as TNF-α, monocyte chemotactic protein-1 (MCP-1), transforming growth factor-β (TGF-β), and tissue inhibitors of metalloproteinase, leading to fibrosis of the liver [127]. Moreover, chemokines and chemokine receptors involved in leukocyte recruitment such as CXCL8/CXCR1; CXCL1 & 3/CXCR2; CCL3-5/CCR5 and the chemokines CXCL9-11 and CCL2 (MCP1) are upregulated in steatohepatitis [128]. Importantly, bacteria toxins, cytokine and chemokines activate JNK and JNK activation is required for synthesis of proinflammatory cytokines [129]. Blood lymphocytes, monocytes and macrophages, and tissue residence immune cells express JNK, SAB and MAP kinase cascades ubiquitously [130]. Therefore, JNK-SAB-ROS activation loop may play a role in sustained activation of P-JNK, and immune cell function and migration are regulated by P-JNK activity. Deletion of SAB or inhibition of JNK kinase activity by SP600125 prevented TNF induced sustained P-JNK activation and cell death [52]. Consistently, conditional mononuclear cell JNK KO mice are prevented from diet-induced NASH [131]. However, a recent study on the differential role of recruited monocytes in steatohepatitis has been discovered [132] and role of MAP kinase signaling in subpopulation of macrophage needs to be revisited.

In addition, CD62E (E-Selectin) and CD44 which recruit leukocytes into inflammation sites are upregulated in NASH patients [128]. P-selectin produced from platelets and endothelial cells by TNF, IL1, or LPS upregulate E-selectin expression in liver sinusoidal endothelial cell (LSEC) [133]. Similar to hepatocytes, LSECs express MAP kinase cascade and SAB, and lipotoxic stress also occurs in LSECs contributing to NOX1 expression and ROS generation [134]. The gut microbiota also seems to contribute to liver endothelial dysfunction. Restoration of a healthy microbiota via fecal transplantation normalizes portal hypertension by improving intrahepatic vascular resistance and endothelial dysfunction in rats [135]. Gut microbiome dysregulation (dysbiosis) is one of the important factors in the progression of NAFLD to advanced fibrosis and cirrhosis [84]. A low fiber, high fat, and high carbohydrate diet changes gut microbiome which in turn changes gut and systemic dietary metabolites such as acetate and cytokines. Gut bacteria and their metabolites translocate to the liver through a disrupted gut barrier and induce hepatic inflammatory reaction [84]. Further diet induced NASH studies are required to explore the effect of microbiome on JNK activation loop in LSEC.

JNK activation in HSCs in response to TGF-β and platelet-derived growth factor (PDGF) activates Smad2/3 leading to α-smooth muscle actin (αSMA) expression, migration of resident HSCs and myofibroblasts, and fibrosis in NASH [17]. Notably, follistatin like 1 (Fstl1), glycoprotein, is secreted from HSCs/myofibroblast induced by TGF-β1. Fstl1 binds to TGF-β1 and negatively regulates TGF-β1 signaling in lung development. Interestingly, Fstl1 neutralizing antibody attenuates CCL4 induced liver fibrosis [136]. Further studies are required to reconcile results from developmental and disease models. Importantly, we found that JNK, SAB and MAP kinase cascade express in HSC, but how JNK activation is involved in HSC function and liver fibrosis is yet to be examined.

## 4. Perspective of the Sustained JNK Activation Loop in Therapeutic Development

Chronic hepatic inflammation causes fibrosis leading to liver cirrhosis. One-carbon metabolism, mitochondrial stress, ER stress, DNA damage, inflammation, and obesity are involved in carcinogenesis in NAFLD. Patients who develop cirrhosis related to NASH are at risk for hepatocellular carcinoma and/or end-stage liver failure and liver transplantation. Currently there is no drug approved by US Food and Drug Administration (FDA) for the treatment of NASH. Lifestyle alterations remain the only treatment. Steatosis, inflammation, ballooning, and fibrosis were improved in those achieving >5% weight loss, with even greater improvement in patients achieving >10% weight loss. Several molecules are in clinical trials (www.clinicaltrials.gov, accessed on 18 July 2022). However, the study that aims to reduce level of P-JNK activation via targeting sustained-JNK-activation-loop is very sparse and mentioned here, and prospective targets are discussed.

**Oxidative stress and antioxidant supplement:** Deletion of Nrf2 results in rapid progression of NASH [137]. In addition, NADPH oxidases (NOXs), which links NAFL progression to NASH and HCC, are membrane-bound enzymatic complexes generating ROS and are abundant in liver associated with inflammation and immune responses [138]. Evidence supports that in early steatosis phase of diet induced NASH oxidized protein adducts increase in liver [78], suggesting that ROS produced in both hepatocytes and non-hepatocytes contribute the progression of NASH. Nevertheless, antioxidants, such as vitamin E prevent NASH [139]. The effectiveness of Vitamin E supplementation is currently under study in clinical trials of NAFLD. Further studies may be indicated because of possible association of prostate cancer and insulin resistance with the long-term usage of Vitamin E [140].

**JNK and JNK inhibitors:** JNK inhibitors inhibit the kinase activity of JNK isoforms in both hepatocytes and non-hepatocytes. SP600125, the most well tested JNK inhibitor, inhibits JNK kinase activity via inhibition of self-activation or MKK4 by competitive reversible inhibition at ATP binding site, but not at a JNK substrate binding site [141]. New and more specific JNK inhibitors, JNK-IN-8 and JNK-IN-10, are irreversible inhibitors [142]. Chemical inhibitors may potentially target a wide variety of JNK substrates in hepatocytes, KCs, HSCs, sinusoidal endothelial cells, and immune cells in liver. This non-selective inhibition precludes the development of JNK inhibitors as drugs for clinical use. The antisense oligo nucleotide targeting to JNK (JNK-ASO) to reduce JNKs expression is a selective and potential therapeutic agent and can efficiently target to hepatocytes [143]. However, hepatic JNK1/2 embryonic KO studies demonstrated that hepatocyte proliferation at 48 h is reduced in an experimental model of liver regeneration [144], though overall regeneration after 72 h is not different. Importantly, prolonged hepatic JNK deficiency increased cholangiocyte proliferation and intrahepatic cholangiocarcinoma [125].

**JNK-SAB interaction and targeting SAB:** SAB is a mitochondrial outer membrane protein. The interaction of P-JNK and SAB inhibits the mitochondrial respiration [51]. SAB expression is increased in human and murine NAFL and NASH [78]. Increased level of SAB amplifies and sustains JNK activation via the JNK-SAB-ROS activation loop. Blocking JNK-SAB interaction by a SAB peptide selectively prevents the sustained amplification of P-JNK and cell death in various animal models [51,52,53,54,55,56,57,58,59,78]. SAB-ASO treatment, which decrease SAB expression and JNK activation, effectively prevents NASH progression and reverses NASH score [78]. Notably, complete depletion of SAB prevents sustained P-JNK activation but does not interfere the initial activation of JNK and stress response signaling required for cellular response and survival such as JNK/ATF2 dependent early phase induction of DUSP1 and 10, and late phase transcriptional activation of DUSP4 and 16 [30].

**JNK-JIP1 interaction and inhibitor:** JNK-interacting protein-1 (JIP1) is a scaffolding protein where upstream MAPKs activate JNK. BI-78D3 is able to compete with the D-domain of JIP1 for JNK binding and thus inhibits JNK activation. BI-78D3 effectively prevent CCL4 induced acute liver injury [145], yet the effect of BI-78D3 in NASH models need to be examined.

**ASK1 and ASK1 inhibitor:** Apoptosis signaling-regulating kinase 1 (ASK1) activity directly involves P-JNK-SAB-ROS activation loop through redox sensitive thioredoxin [146,147]. ASK1 is a MAP3 kinase which activates downstream terminal kinases both JNK and p38. JNK activity on mitochondria reversely corresponds to mitochondrial respiration [51], but function of p38 on mitochondria is not known. ASK1 inhibitor, Selonsertib (GS-444217) ameliorates NASH and improved fibrosis in preclinical studies and in a short-term clinical trial [148] but was not effective in late-phase clinical trials [149]. Selonsertib, which blocks ASK1 activation but not MLKs, may not be enough to revert the progression of NASH in human disease. Notably, hepatic JNK1 and JNK2 are cross-activated by the upstream MAP3Ks, ASK1, MLK2/3 and TAK1.

**MLKs and MLK inhibitor:** The stress kinase mixed lineage kinase is also a MAP3K activating MKK4/7 and then JNK/p38. MLK3-JNK mediates the hepatic extracellular vesicle release [150,151]. Genetic or pharmacological inhibition of MLK3 results in reduction of the potent C-X-C motif chemokine ligand 10 (CXCL10) in extracellular vesicles (EV) derived from LPC-treated hepatocytes. NASH-inducing diet fed *MLK3^−/−^* mice have reduced CXCL10 levels in their plasma EVs and, hepatoprotection against injury and inflammation. Furthermore, pharmacological MLK3 inhibitor, URMC099, reduces circulating CXCL10 and attenuates murine NASH [151]. It is important to stress that circulating EVs derived from hepatocytes, immune cells and platelets can be a biomarker for NAFL resolution in response to weight loss surgery [152,153]. Preclinical studies of URMC099 on Alzheimer’s disease, Parkinson’s disease and NASH model are encouraging [154,155]. Efficacy in human clinical trials has not yet been examined.

## 5. Conclusions

The understanding the mechanism of kinase activation and signal regulation will generate therapeutic target molecules to develop a safe and effective treatment. However, limitations and caveats do exist because of similarity of the kinase activation and interaction with its substrates through a pattern of amino acid sequence, motif, creating unwanted off target side effects. Finding the targetable proximity molecules in the P-JNK activation loop, Table 1 such as CFLAR in ASK1 activation, could be an alternative strategy. Advances in our current understanding of molecular mechanisms using improved animal models and small molecule screening could support the new pharmaceutical development especially target specific peptides or antisense oligos, which are the promising technologies for the future prevention and treatment of NASH.

## Figures and Tables

**Figure 1 biomedicines-10-02035-f001:**
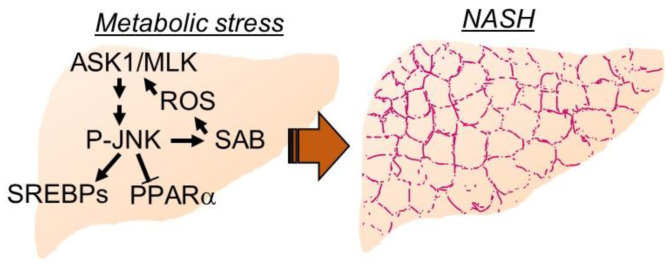
**Key mediators of hepatic metabolic stress.** Carbohydrate and free fatty acid overload to hepatocytes activate stress kinase cascade to upregulate *de novo* lipogenesis genes to adapt metabolic stress. Dial-up feedforward activation of stress kinase cascade through P-JNK-SAB interaction attenuates β-oxidation and lipid oxidation genes. Damage signals, receptors, and extracellular vesicles from hepatocytes recruit inflammation, and activate hepatic stellate cells and fibrogenesis.

**Figure 2 biomedicines-10-02035-f002:**
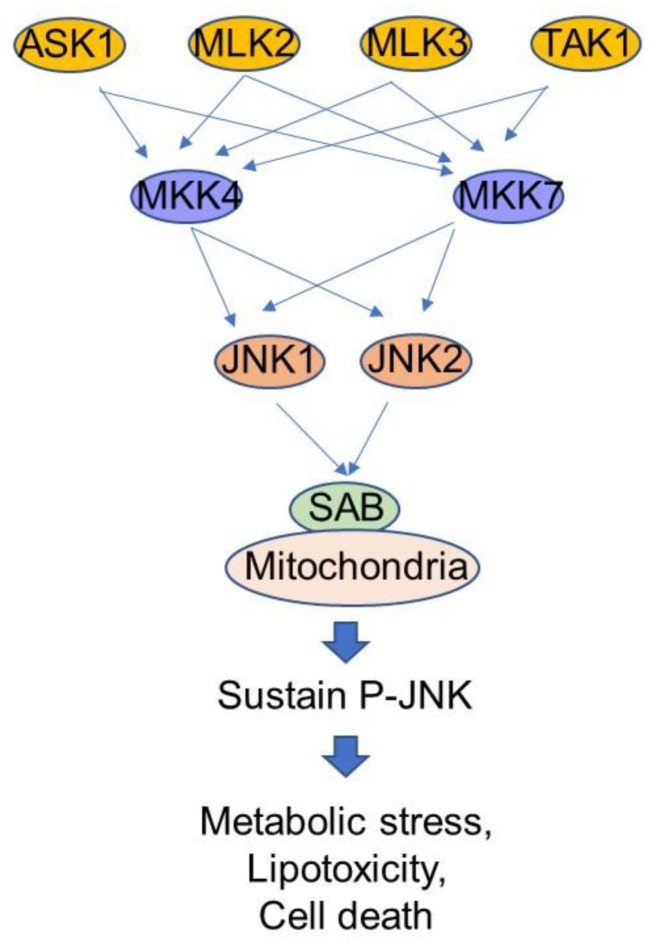
**Stress response kinase cascade.** MAP3K such as ASK1, MLK2/3, TAK1 are upstream kinases which are activated by reactive oxygen species, membrane lipid composition and changes. MKK4 and MKK7 are abundant MAP2K in liver. JNK1 and JNK2 are MAP kinases with functional redundancy in liver. SAB is a mitochondrial outer membrane protein and directly interacts with P-JNK, but not interact with p38 in vivo. Suppression of SAB expression or inhibition of P-JNK-SAB interaction is a plausible therapeutic target of hepatic metabolic stress in NASH.

**Table 1 biomedicines-10-02035-t001:** **Regulators of JNK activation in liver diseases.** MAPKs, JNK and p38, are activated by upstream MAP kinase cascade which activity is dampened by interacting proteins or proteasomal degradation, or enhanced by membrane associated signal activation or reactive oxygen species (ROS). DUSP family phosphatases inhibition of MAP kinase cascade is critical in preventing susceptibility of hepatocyte toxicity and steatosis. Sustained activation of JNK via SAB-ROS-MAP3K overcomes the cellular protective mechanism and causes hepatocyte toxicity and lipogenesis.

Regulators of JNK Activation in Hepatocyte Toxicity and Diet Induced NASH		
**Kinases**	**Proximity molecules in the context of study**	**Reference**
ASK1	CASP8, CFLAR, TRX, TRAF1, TNFAIP3	[60,85,86]
MLK2/3, LZK	JIP1	[69,71,72,89,149,150]
TAK1	TRAF3, NF-κB, TGFβ,	[49,155,156,157,158]
MKK4/7	Ulk1/2, JIP1,LZK	[60,63,149,159]
JNK	SAB	[11,13,30,52,54,61,78,79,160]
**Phosphatase**	**Target molecules**	
DUSP9, 12	ASK1, JNK, p38,	[44,45]
DUSP10	JNK, p38, Erk	[46,49]
DUSP14, 26	TAK1	[47,48,49]

## Data Availability

The study did not report any data.

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
