# Peer review of "Mechanism and Therapeutic Targets of c-Jun-N-Terminal Kinases Activation in Nonalcoholic Fatty Liver Disease"

_biomedicines, 2022, doi:10.3390/biomedicines10082035_

Round 1
Reviewer 1 Report
Summary: This review discusses the role of mitogen-activated kinases (MAPK), especially c-JUN-N-terminal kinase (JNKs) activation, in the development of non-alcoholic fatty liver disease (NAFLD) and no-alcoholic steatohepatitis (NASH). Furthermore, these pathways are discussed in application as potential targets for NAFLD/NASH therapeutics.
Major Comments:
1. Citations in the text are sparse. Portions of the text, such as Section 2 “Hepatic MAPK family” only cite two references at the very end, so it is unclear where the information in each sentence comes from.
Section 3.5 “Ethnicity, genetic polymorphism, and obesity” only includes a single reference for a relatively broad topic. Furthermore, the included citation appears to be for a single study, so it is unclear where the authors are getting their information for the genetic factors, demographics, and the meta-analysis discussed. Uncited information can be found across this review, including but not limited to:
a. the second half of section 1, the introduction
b. Most of the first paragraph of section 4, “Pharmaceutical target molecules”, especially information concerning the connection between NAFL and Alzheimer’s or Parkinson’s disease
c. “Anti-oxidants” subsection in section 4
d. “Targeting JNK-JIP1 interaction” subsection in section 4
e. “Targeting MAP3K ASK1” subsection in section 4
f. In section 3.6 “Hormones and sexual dimorphism”, the authors mention “clinical and epidemiological studies” that have shown the effect of hormone therapy on NAFLD on postmenopausal women. These original articles should be cited.
g. Section 3.6, lines 11-13: “In addition, skeletal muscle, adipose tissue, and thyroid hormones contribute to metabolic homeostasis and involves in protection from NASH by female sex hormones.” Is uncited and should be rewritten for clarity.
h. Section 4, paragraph 2: citations are needed for claims concerning current treatments, and the effect of weight loss on NASH.
Sections like 3.1 “Hepatic steatosis and oxidative stress” do appear carefully and completely cited. The manuscript should be more consistent with this section. Additional references could be included to better support the thoroughness of their review.
2. In a mouse study of NAFLD and fibrosis, Challa, et al. [https://doi.org/10.15252/emmm.201810124] report that ASK1 acts as a suppressor of NASH and fibrosis. Specifically, in studies of ASK1-knockout mice and pharmaceutical inhibition of ASK1 accumulation of hepatic lipids were observed, while overexpression of liver ASK1 reduced steatosis and fibrosis. This potential protective role of ASK1 contradicts information in this review. How do these findings fit into the context of this review? The authors should cite this work, and should look for additional literature concerning more nuanced effects of MAPKs in steatosis/fibrosis.
3. The first paragraph of section 4 “Pharmaceutical target molecules” discusses other diseases associated with NAFLD and NASH. While this is interesting, this text seems out of place in this section. The authors should consider creating a new section to discuss comorbidities, or this discussion should be moved to the introduction. As mentioned in the previous comment, most of this paragraph also requires citations for the information included in this text. Additionally, the authors should edit this paragraph for improved fluency.
4. This manuscript would be greatly benefited by an additional table which summarizes the pharmaceuticals and clinical trials discussed in section 4. Specifically, a table that included information such as the drug name, target, animal testing or clinical trial phase, result, as well as a specific reference to this information like the clinicaltrials.gov identifier would provide a clear summary of the state of pharmaceutical development for NAFLD.
5. The conclusion should be revised. Specifically, the authors mention the Koli diet and DIAMOND model as mouse models of NAFLD/NASH, and are not defined or discussed anywhere else in the text. While discussing the importance of improved animal models for future pharmaceutical development does seem like an important point, this statement seems out of context with the rest of the manuscript as is.
If the authors believe this is an important comment for the state of the field, a subsection on current animal models of MAPK activation, NAFLD/NASH, and their short comings should be added. A table including this information could also be helpful.
Considering the specific focus on MAPK activation in NAFLD, the conclusion should include statements on the state of what is known and areas that need further study. There are various interesting gaps of knowledge identified in section 3 that could be reiterated. Moreover, a comment on the state of pharmaceutical development is needed. Is there a promising drug candidate? If not, is there something about the MAPK/JNK activation pathway that should be further considered in the development of new drugs?
Minor Comments:
1. The authors are inconsistent with their use of “NAFL” or “NAFLD” as an acronym for non-alcoholic fatty liver/non-alcoholic fatty liver disease.
2. Beyond the effect of changes in sex hormones (discussed in section 3.6), is there a known effect of age on MAPK/JNK activation in NAFLD/NASH?
3. There are a few minor grammatical errors which should be fixed:
a. Line 18 of the introduction, “MAPKS” used instead of “MAPKs”
b. In line 4 of section 3.1 “Hepatic steatosis and oxidative stress”, “…steatotic/steatohepatitis” might make more sense written as “…steatosis/steatohepatitis.”
c. In lines 2-3 of section 3.3 “Gut microbiota”, edit “A low fiber and high fat high carbohydrate diet…” to “A low fiber, high fat, and high carbohydrate diet...”
Author Response
Detailed response to the Editor’s and Reviewers’ comments:
We appreciate Editor and Reviewers very much again for the constructive comments and the opportunity to revise and improve our manuscript. We have addressed ALL the issues that were raised.
We changed the title to match the review context. The new title is “Mechanism and Therapeutic Targets of c-Jun-N-Terminal Kinases Activation in Nonalcoholic Fatty Liver Disease”.
To editor
The title of the review manuscript is changed to “Mechanism and Therapeutic Targets of c-Jun-N-Terminal Kinases Activation in Nonalcoholic Fatty Liver Disease” to focus on mechanism and application of sustained JNK activation loop.
Reviewer #1
- Citations in the text are sparse.
Authors’ response: Thank you for the suggestion. We added relevant citations throughout the manuscript.
- In a mouse study of NAFLD and fibrosis, Challa, et al.report that ASK1 acts as a suppressor of NASH and fibrosis.
Authors’ response: Thank you. We added Challa, et al.’s report. Challa, et al. reported compelling evidence of ASK1 role in protection from NASH. However other groups found ASK1 role in progression to NASH. We think there may be systematic differences in materials and methods used between various laboratory such as method and generation of flox mice, background strain, food content, housing environments. This controversy should be resolved and the discussion is opened until further evidences.
- The first paragraph of section 4 “Pharmaceutical target molecules” discusses other diseases ……………
Authors’ response: We have rewritten and rearranged the manuscript to focus on mechanism and application of sustained JNK activation loop.
- This manuscript would be greatly benefited by an additional table which summarizes the……
Authors’ response: Thank you for suggestion to present with Table for pharmaceutical drug lines used in NASH. This review manuscript is to deliver important of “sustained JNK activation loop also known as JNK-SAB-ROS activation loop” and therapeutic application. So far only a few drugs such as ASK1 inhibitor selonsertib and vitamin E are approved for clinical trials and not a huge list of drugs to present in a table.
- The conclusion should be revised.
Authors’ response: Animal models are not relevant in this review and removed. Knowledge gaps are stated at the end of appropriate paragraph.
Reviewer 2 Report
This is an interesting review on the role of MAP kinases in the pathogenesis of NAFL with possible therapeutic implications. However there are certain flaws that limit the presentation of the subject.
1) The authors tried to refer to many aspects of the complex pathogenesis of NAFL and NASH. The result is a superficial review of the problem and lack of clarity on the role of the kinases. It would be preferable to focus only on the mechanisms implicating MAP kinases having clearly explaining their role in liver homeostasis for the non expert reader.
2) The drawback is exemplified by the fact that the role of autophagy which is critical in the pathogenesis of NAFL is mentioned as a simple word at the conclusion.
3) Relevant to these comments is the fact that there are many statements without the proper literature support. For example the whole section 4 contains only 4 references despite the fact that it refers to the main point of the review, the therapeutic implications.
4) Minor point: In the introduction the authors state that the majority of patients with NAFL require an effective treatment to prevent progression to NASH. Yet in the previous paragraph they state that only 20% of NAFL patients will progress to NASH.
5) The Use of English require attention in syntax.
Author Response
Detailed response to the Editor’s and Reviewers’ comments:
We appreciate Editor and Reviewers very much again for the constructive comments and the opportunity to revise and improve our manuscript. We have addressed ALL the issues that were raised.
We changed the title to match the review context. The new title is “Mechanism and Therapeutic Targets of c-Jun-N-Terminal Kinases Activation in Nonalcoholic Fatty Liver Disease”.
To editor
The title of the review manuscript is changed to “Mechanism and Therapeutic Targets of c-Jun-N-Terminal Kinases Activation in Nonalcoholic Fatty Liver Disease” to focus on mechanism and application of sustained JNK activation loop.
Reviewer #2
Author’s responses are
- More detail mechanism of sustained activation of JNK and MAPK cascade regulation is described in this revision with references for application of other area of research.
- Thank you for suggestion on autophagy. Autophagy specifically lipophagy is one of the important mechanisms contributing to prevent NASH, and included in the manuscript. However, there is not publications to discuss role of JNK activation in mechanism of autophagy. Autophagy is highest at fasting, and P-JNK in liver is lowest at fasting. We need further evidence to describe involvement of JNK in mechanism of autophagy.
- Supportive literatures are added.
Round 2
Reviewer 2 Report
The paper has been extensively re-written. Most confusing irrelevant details have been removed and the molecular pathways have been clarified and comprehensively discussed. References have been adequately added.